# Does Forward Head Posture Influence Somatosensory Evoked Potentials and Somatosensory Processing in Asymptomatic Young Adults?

**DOI:** 10.3390/jcm12093217

**Published:** 2023-04-29

**Authors:** Ibrahim M. Moustafa, Aliaa Attiah Mohamed Diab, Deed E. Harrison

**Affiliations:** 1Department of Physiotherapy, College of Health Sciences, University of Sharjah, Sharjah 27272, United Arab Emirates; 2Neuromusculoskeletal Rehabilitation Research Group, RIMHS—Research Institute of Medical and Health Sciences, University of Sharjah, Sharjah 27272, United Arab Emirates; 3Faculty of Physical Therapy, Cairo University, Giza 12613, Egypt; 4CBP Nonprofit (A Spine Research Foundation), Eagle, ID 83616, USA

**Keywords:** forward head posture, cervical spine, somatosensory evoked potential

## Abstract

The current investigation used somatosensory evoked potentials (SEPs) to assess differences in sensorimotor integration and somatosensory processing variables between asymptomatic individuals with and without forward head posture (FHP). We assessed different neural regions of the somatosensory pathway, including the amplitudes of the peripheral N9, spinal N13, brainstem P14, peak-to-peak amplitudes of parietal N20 and P27, and frontal N30 potentials. Central conduction time (N13–N20) was measured as the difference in peak latencies of N13 and N20. We measured these variables in 60 participants with FHP defined as a craniovertebral angle (CVA) < 50° and 60 control participants matched for age, gender, and body mass index (BMI) with normal FHP defined as CVA > 55°. Differences in variable measures were examined using the parametric *t*-test. Pearson’s correlation was used to evaluate the relationship between the CVA and sensorimotor integration and SEP measurements. A generalized linear model (GLM) was used to compare the SEP measures between groups, with adjustment for educational level, marital status, BMI, and working hours per week. There were statistically significant differences between the FHP group and control group for all sensorimotor integration and SEP processing variables, including the amplitudes of spinal N13 (*p* < 0.005), brainstem P14 (*p* < 0.005), peak-to-peak amplitudes of parietal N20 and P27 (*p* < 0.005), frontal N30 potentials (*p* < 0.005), and the conduction time N13–N20 (*p* = 0.004). The CVA significantly correlated with all measured neurophysiological variables indicating that as FHP increased, sensorimotor integration and SEP processing became less efficient. FHP group correlations were: N9 (r = −0.44, *p* < 0.001); N13 (r = −0.67, *p* < 0.001); P14 (r = −0.58, *p* < 0.001); N20 (r = −0.49, *p* = 0.001); P27 (r = −0.58, *p* < 0.001); N30 potentials (r = −0.64, *p* < 0.001); and N13–N20 (r = −0.61, *p* < 0.001). GLM identified that increased working hours adversely affected the SEP measures (*p* < 0.005), while each 1° increase in the CVA was associated with improved SEP amplitudes and more efficient central conduction time (N13–N20; *p* < 0.005). Less efficient sensorimotor integration and SEP processing may be related to previous scientific reports of altered sensorimotor control and athletic skill measures in populations with FHP. Future investigations should seek to replicate our findings in different spine disorders and symptomatic populations in an effort to understand how improving forward head posture might benefit functional outcomes of patient care.

## 1. Introduction

Sensorimotor integration and central somatosensory processing are brain processes that allow for the execution of certain voluntary motor behaviors in response to specific demands of the environment [1]. In other words, it is the synergistic relationship between the sensory and motor systems [2]. Thus, the behavior pattern of healthy individuals and movement disorder patients depends on the sensorimotor integration process [3]. Alterations in sensorimotor integration and somatosensory processing may offer insights into differences in patient motor control abnormalities and disturbances seen in specific spinal disorders with neurologic components [4,5].

Chronic pain is a strong contributing factor triggering sensorimotor integration alterations [6,7,8]. It is known to alter specific regions of the brain functionally and structurally, such as the amygdala, the anterior cingulate cortex, the medial prefrontal cortex, and the primary somatosensory cortex. These alterations are considered maladaptive as they result in hyper-excitability and pathway re-organization [9,10]. Theoretically, altered afferent input is a likely explanation for the production and sustained occurrence of central neurophysiological processing dysfunctions [11,12,13]. The primary motor cortex (termed M1) is considered the central station where sensory input from the peripheral systems converges and is processed in order to execute proper and efficient voluntary motor tasks (sensorimotor integration). Sensorimotor integration also occurs in other regions of the brain (the parietal cortex, the supplementary motor area, the dorsal premotor cortex, the ventral premotor cortex, the basal ganglia, the cerebellum, and the thalamus, to name a few). These regions are known to alter and/or contribute to voluntary motor tasks as well. In simple terms, abnormalities of the peripheral (extremity) and central (spinal) tissues responsible for contributing to sensory input into the sensorimotor integration and somatosensory processing systems can cause disruption or dysfunction in the normal afferent input and processing in the M1 region, and thus, lead to inefficient motor control output [14,15].

There are many important questions regarding sensorimotor integration and somatosensory processing remaining to be addressed. For instance, the relevance of altered alignment of the sagittal cervical spine in symptomatic and asymptomatic persons to function/dysfunction in the sensorimotor integration and somatosensory processing systems remains understudied. It is known that the magnitude of forward head posture (FHP) is inversely correlated to the cervical spine range of motion [16]. Furthermore, FHP alters the length of the cervical spine through kinematic flexion/extension coupling and alters load sharing among the discs, ligaments, and muscles of the cervical spine [17,18]. Investigations on sustained cervical spine flexion have found changes in afferentation and abnormal feed-forward control due to mechanical viscoelastic changes to the cervical spine soft tissues that affect position sense repeatability [19]. Furthermore, straightening of the cervical spine lordotic curvature (as often occurs with FHP) has been found to significantly reduce the F-wave in the median nerve of the upper limbs of tested individuals, indicating a reduction in motor–Neuronal excitability [20]. Relatively few studies have addressed the relationship between FHP and inefficient sensorimotor integration and somatosensory processing [21,22].

Therefore, the purpose of the current investigation is to compare the sensorimotor integration and somatosensory processing at different neural regions of the somatosensory system, including central conduction time, in persons with and without forward head posture (FHP) and without overt symptomatology. The specific research questions to be addressed herein include: (1) Using somatosensory evoked potentials, is there a difference in sensorimotor integration and processing in asymptomatic participants without FHP compared to participants with FHP?; (2) Do persons with FHP have abnormal sensorimotor integration and at what region(s) does this occur?; (3) Is the possible alteration to somatosensory processing linearly related to the amount of FHP displacement?

## 2. Materials and Methods

Participants were collected as a convenience sample of asymptomatic individuals. Recruitment was obtained using both printed advertisements and social media. These advertisements were directed only to university-related communities, such as employees, alums, and students. All the participants were asymptomatic and had not received any physical therapy or any type of manual therapy treatment in the last year between November 2021 and July 2022. Ethics approval was obtained from our University (College of Health Sciences, University of Sharjah, UAE) (Ethical approval number: REC-21-03-11-03-S), and informed consent was provided to and obtained from all participants prior to data collection, in accordance with relevant guidelines and regulations.

### 2.1. Participants

Sixty participants with definite forward head posture (FHP) and sixty matched control participants without FHP were recruited for this study. Participants were matched for age, sex, demographics, and body mass index (BMI). In order for a participant to be categorized as having FHP, the craniovertebral angle (CVA) measurement was used, and published cutoff values were followed. Utilizing the data published by Yip et al. [23], FHP was classified as having a CVA < 50°; thus, participants were in the FHP group when CVA was <50°. Conversely, the control group was defined as having normal or no FHP when a participant’s CVA was >55°. All FHP screening procedures were carried out by a physiotherapist with 15 years of clinical experience.

As standard practice, clinicians with 10 years of experience assessed all participants. Exclusion criteria for the current investigation were as follows: (i) any inflammatory joint disease; (ii) any systemic pathology; (iii) a history of significant injury or primary musculo-skeletal surgical interventions; (iv) deformity of the spine or extremities; and (v) any pain in the past 3-months involving the musculo-skeletal system. All participants were required to be pain-free. This was done in order to assess the potential effects of abnormal head posture without the presence of acute pain, as the presence of pain alone is known to induce a significant reduction in the post-central N20–P25 complex and a significant increase in the N18 wave [24].

### 2.2. Measurement Techniques

#### 2.2.1. Craniovertebral Angle (CVA)

The CVA is reliable and valid for the assessment of FHP [25]. The CVA is measured as the angle of intersection between a horizontal line and a line bisecting the tragus of the ear and the C7 spinous process. We followed a previously published protocol for the measurement of the CVA in a neutral, relaxed sitting position [26]. Lateral photographs of each participant were taken with the instructions for them to be seated in a comfortable, relaxed, and neutral position. A tripod, with a mounted digital camera positioned 0.8 m from the sitting participant, was placed perpendicular to the sagittal plane of the participant. The height of the camera was set at the height of each person’s seventh cervical vertebra. To identify the tragus of the ear and the 7th cervical spinous process, adhesive markers were fixed on these two landmarks, which then allowed the measurement of the CVA on the photographs. Figure 1 depicts the CVA measurement used with a representative participant with (a) normal head posture and (b) considerable forward head posture (FHP).

#### 2.2.2. Evaluation of Sensorimotor Integration and Somatosensory Processing

Sensorimotor integration and somatosensory processing were assessed using the neurophysiological measured variables, including amplitudes of the following potentials: the peripheral N9; spinal N13; brainstem P14; parietal N20 and P27; and frontal N30. Differences in peak latencies between N13 and N20 were measured as the central conduction time (N13–N20). In order to assess the neurophysiological variables in this study, we used an electromyogram device (Neuropack S1 MEB-9400K, Nihon Koden, Japan). We followed the protocol previously reported in our earlier investigation and repeated key components herein for clarity of understanding [21]. The skin was cleaned, and the stimulating electrodes were placed on the skin overlying the median nerve 2–3 cm superiorly relative to the distal crease of the wrist. We used a bearable, painless stimulus intensity set at 3 times above the sensory level. No participant reported this as noxious or pain-causing [21].

For recording, all somatosensory evoked potential (SEP) recording electrodes (7 mm Ag-AgCl disposable adhesive electrodes from Neurosoft) were placed according to the International Federation of Clinical Neurophysiologists’ (IFCN) recommendations [21,27]. Careful attention was paid to cleaning and scarifying the skin before the attachment of the recording electrodes on the scalp. Using an impedance below 5 kΩ, recording electrodes were placed over the ipsilateral Erb’s point, superficial to the sixth cervical vertebra spinous process (Cv6). Additional recording electrodes were placed at the frontal and parietal scalp regions contralateral to the side of stimulation at 2 cm posterior to the contralateral central and frontal scalp cites C3/4 and F3/4, which are referred to as Cc′, and Fc′, respectively. Frontal and partial recording electrodes were referenced to the ipsilateral earlobe [27]. The C6 spinous electrode was referenced to the anterior neck (tracheal cartilage). The Erb’s point electrode was also referenced to the contralateral shoulder, as SEP components originating from subcortical regions are best recorded with a non-cephalic reference [21,28]. A ground electrode was attached to the forehead FPz. Figure 2 demonstrates this procedural setup.

Our study protocol utilized previously published protocols [29,30,31]. The band was set between 5 and 1500 Hz, with a time of 100 ms and a bandwidth of 103 μs. Using an electrical square pulse stimulus with a duration of 0.2 ms, a total of 800 sweeps were performed and averaged. We repeated each test a minimum of two times, where the summated tracings were quantified for the amplitude and latency of the potentials [29,30,31]. The amplitude of the individual SEP components was measured from their peak to the preceding or succeeding trough according to the IFCN guidelines [27]. The following potentials were assessed and recorded:The peripheral N9;The spinal N13 potential to the succeeding positive trough [21,31];The far-field P14–N18 complex [21];The parietal N20 (P14–N20 and N20–P27 complexes) [32];The frontal N30 (P22–N30 complex) [21,33]. The N30 potential reflects the functional connectivity of sensorimotor integration, which includes the thalamus, premotor area, basal ganglia, and primary motor cortex [33,34,35,36].

The amplitude of each respective peak represents the degree of activity of its neural structure. Alterations are believed to reflect alterations in the amount of activity of the same assumed neural structures [27]. Peak-to-peak amplitude potentials were measured. We used two different rates to process the different potentials: (1) the slower rate of 2.47 Hz was optimum for N30, while (2) a faster rate of 4.98 Hz was used to quantify the potentials for N13, P14, N20, and P27.

To assess central conduction time (N13–N20), median nerve stimulation at the wrist of each participant was performed and determined [37,38]. Differences in peak latencies between N13 and N20 waves function as a measure of the conduction time along the central and spinal somatosensory pathways. All neurophysiological measures were carried out by a physiotherapist with 20 years of experience in such measurement techniques. All measurements were conducted at the EMG research laboratory, University of Sharjah, UAE.

### 2.3. Sample Size Determination

We used data from our previous study [21] to estimate the sample size needed to identify differences in somatosensory integration measures between participants with and without FHP. The mean differences and standard deviation of the N30 potential were estimated to be 0.5 and 0.6, respectively, from this study. Accordingly, at least 60 participants per group, given a significance level of 5% and a statistical power of 80%, were needed in the current study [21].

### 2.4. Data Analysis

The normal distribution of all descriptive baseline variables was determined using the Kolmogorov–Smirnov test, where continuous data are noted as mean with standard deviation (SD) in the text and tables. Equality of variance was assessed with Levene’s test, attaining a 95% confidence level, *p*-value < 0.05. Descriptive statistics (means ± SD unless otherwise stated) are listed at each time point. In order to identify if group equivalence was achieved for proper case-control analysis, a Student’s *t*-test for continuous variables or Chi-squared for categorical variables test was performed for each demographic and clinical variable [21].

The Student’s *t*-test was used to compare the means of continuous variables between the two groups. A *p*-value of less than 0.05 was considered statistically significant. The effect size was calculated using Cohen’s d where d ≈ 0.2 indicates negligible clinical importance, d ≈ 0.5 indicates moderate clinical importance, and d ≈ 0.8 indicates high clinical importance [39]. Correlations (Pearson’s *r*) were used to examine the relationships between the CVA (in the study and control groups) and the measured variables: amplitudes of the peripheral N9; spinal N13; brainstem P14; parietal N20 and P27; frontal N30 potentials; and the central somatosensory conduction time (N13–N20).

A generalized linear model was used to compare the neurophysiological scores between groups, with adjustment for potential confounding variables (educational level, marital status, BMI, and number of working hours per week). Multiple logistic regression models were used to assess the predictors of the neurophysiological outcomes (P14, N20, P27, N30, N13, and N13–N20). SPSS version 20.0 software was used for analyzing data (SPSS Inc., Chicago, IL) with normality and equal variance assumptions ensured before the analysis [21].

## 3. Results

Initially, 680 potential participants were screened. Neck pain and shoulder pain were the most common reasons for participant exclusion. Sixty participants with FHP (mean age 23.5 years, SD = 2; 35 males, 25 females) and sixty age-, BMI-, and sex-matched controls without FHP were recruited. Figure 3 shows the participant flow chart.

### 3.1. Demographic Characteristics of the Participants

Descriptive data for baseline participant demographics are presented in Table 1. No statistically significant differences between the control and the FHP groups were found at baseline in any of their demographic variables; *p* > 0.05. The mean and distribution of craniovertebral angle for both groups are shown in Figure 4.

### 3.2. Between Group Analysis

Statistically significant differences between the FHP and control groups for all measured neurophysiological variables were identified, including amplitude of spinal N13 (*p* < 0.005), brainstem P14 (*p* < 0.005), parietal N20 and P27 (*p* < 0.005), frontal N30 (*p* < 0.005), and N13–N20 interpeak latency as a measure of central conduction time (CCT) (*p* = 0.004). There was no significant difference between groups regarding the amplitudes of the peripheral potential N9 (*p* = 0.07). The effect size (Cohen’s d) was moderate for only one variable (N13–N20) and of high clinical significance for the remaining variables. Table 2 and Figure 5 report these data. Figure 6 shows an example of the frontal, parietal, and cervical somatosensory findings for a representative participant with (a) normal head posture and (b) considerable forward head posture (FHP).

### 3.3. Correlation of Findings between Groups

For correlation findings, significant negative correlations were identified between the amount of CVA and the measured neurophysiological variables in both groups. Specific to the FHP group the correlations were: amplitudes of the peripheral N9 (r = −0.44, *p* < 0.001); spinal N13 (r = −0.67, *p* < 0.001); brainstem P14 (r = −0.58, *p* < 0.001); parietal N20 (r = −0.49, *p* = 0.001); P27 (r = −0.58, *p* < 0.001); frontal N30 potentials (r = −0.64, *p* < 0.001); and for central conduction time the correlation was N13–N20 (r = −0.61, *p* < 0.001). Table 3 reports these data.

### 3.4. Logistic Regession Modelling

Working hours and the CVA angle measures significantly affected the neurophysiological outcomes. Full-time work significantly increased the odds of having a higher amplitude of the neurophysiological potentials and slower N13–N20 conduction time when compared with part-time work; *p* < 0.005. Additionally, each 1-degree increase in the CVA measurement significantly decreased the amplitudes of all the potentials and resulted in a faster, more efficient N13–N20 conduction time; *p* < 0.005. Table 4 reports these data.

## 4. Discussion

Using somatosensory evoked potentials, we investigated possible differences in sensorimotor integration and somatosensory processing variables between asymptomatic young adults with FHP and a control group with normal head posture. Our findings indicated that forward head posture, as measured with the CVA, has an impact on sensorimotor integration and somatosensory processing parameters. These findings confirmed our study’s hypotheses. We believe this is the first investigation to provide clear evidence that the amount of FHP alignment influences these specific neurophysiological measures in asymptomatic persons. In our between-group analysis, the only non-significant finding (small effect size) was for N9, which reflects the peripheral nerve volley at the axilla. This finding ruled out peripheral nerve entrapment as a possible cause of any change. Using generalized linear modeling with adjustment for confounding variables, working hours per week and the CVA magnitude were found to affect the neurophysiological outcomes significantly. Surprisingly, full-time work was found to increase the odds of having a higher amplitude of the neurophysiological potentials and slower N13–N20 conduction time when compared with part-time work, indicating an adverse effect on somatosensory processing variables herein. In contrast, each 1-degree increase in the CVA measurement (indicating better posture) significantly decreased the amplitudes of all the potentials and resulted in a faster, more efficient N13–N20 conduction time.

### 4.1. Cortical, Subcortical, and Spinal Neural Changes

We identified sensorimotor integration differences and somatosensory processing changes between both groups occurring in different regions of the spinal and cortical regions. Previous investigations have identified results that are generally consistent with our findings [34,35,36,40,41]. Likewise, previous research using symptomatic populations has found that a general abnormal afferentation process is responsible for spinal, cortical, and subcortical reorganization [29,30]. Thus, reorganization of the somatosensory system is primarily driven by alterations to or modifications of sensory input, which, in turn, alters sensorimotor integration and generalized somatosensory processing [11,12,13,42].

The idea that increased and abnormal FHP is a primary mechanism having the ability to alter afferent input leading to disturbances in the sensorimotor and somatosensory processing system, is not without evidence. Sagittal plane cervical biomechanics studies have identified that tissue component (muscle, tendon, disc, bone) stress and strain are increased due to increasing FHP [17,18]. Further, it is known that as FHP increases, there is an influence on altered joint position, kinematics, and dysfunction that may lead to abnormal neurophysiologic afferent information (so-called dysafferentation). Furthermore, studies suggest that increased FHP may result in increased physical demands, resulting in premature and accelerated degenerative changes in the muscles, ligaments, bone, and neural tissues [5,43,44]. Additionally, abnormal head posture is associated with both a reduced range of movement and an altered segmental cervical spine kinematic pattern. Thus, non-neutral sagittal cervical spine alignment could potentially *lead to* altered sensorimotor integration through an altered afferent input from abnormal cervical spine movements, a change in the muscle-tendon length-tension relationships, and altered spine tissue load sharing [16,17,18,19,20,21,22]. This would seem to explain the findings of Moustafa et al. [21], where collegiate athletes with considerable FHP compared to a control group without FHP were found to be less efficient in athletic skill tests in both static and dynamic situations.

### 4.2. Central Somatosensory Conduction Time

The finding of a faster (more efficient) central condition time in the participants with normal head posture (control group) is likely multi-factorial in nature but may be largely explained by two mechanical phenomena: (1) FHP likely increases longitudinal stress and strain in the spinal cord tissues and (2) increased FHP alters and influences respiratory function. Regarding the former, spinal cord biomechanics, it is expected that participants without FHP or more normal posture alignment also have a more normal (deeper) cervical lordosis [5,45]. A proper cervical lordosis and reduced FHP have been found to reduce stress and strain on the spinal cord, brainstem, nerve roots, and cranial nerves 5–12 in both surgical and non-surgical rehabilitation investigations [4,5,46,47,48,49]. Furthermore, more normal FHP is linearly correlated with an increased overall cervical range of motion [16] because it is known that neural axoplasm has thixotropic properties [50]. It seems logical, therefore, that an increased viscosity (driven by impaired motion and increased spinal cord or neural strain) could alter neuronal transport mechanisms.

Likewise, FHP may act to reduce respiratory functions of both inspiration and expiration volume and strength, and thus, the maintenance of a more neutral sagittal head posture is required to prevent these positional respiratory function reductions [51,52]. Furthermore, since abnormal sagittal plane postures cause an increase in stress and strain on both neural and vascular tissues in the cervical spine [46,47], and it is known that neuronal tissues are highly oxygen-energy dependent [53], it is probable that increased neural strain coupled with reductions in respiratory efficiency may be a mechanism subtly impacting oxygenation to the spinal cord, nerve roots, and cerebral areas, leading to the alteration in the sensorimotor integration disturbances identified in our study. Supporting these assertions, there is evidence of an alteration in vertebral artery hemodynamics and cerebral blood flow intensity on MRA due to alterations in sagittal cervical alignment [54,55].

### 4.3. Clinical Implications

While the observed differences in our neurophysiological data in terms of actual numerical differences can be arbitrary and should not be construed as rigorous in isolation, relating Cohen’s d between our two groups (as in Table 3) to other existing reports in the literature offers context to the meaning or implications of our findings. Of interest, the mean difference and effect sizes for central somatosensory conduction time (N13–N20), sensorimotor integration (N30 potential), and somatosensory processing potentials (spinal N13, brainstem P14, parietal N20 and P27) found in the current study are very similar to the mean differences and effect sizes reported in a previous experimental study [56]. In relation to clinical interpretation, it is thought that alterations in normal afferentation may influence the processing of neural networks located in cortical motor areas and, in turn, impact motor control [14,15]. In support of this concept, it has been identified that collegiate athletes with increased FHP exhibited altered sensorimotor processing, integration, and concomitantly. They were found to have less efficient athletic performance compared to athletes with normal sagittal head posture alignment [21]. Furthermore, a recent randomized trial demonstrated that structural rehabilitation (correction of abnormal alignment) of the sagittal cervical spine allowed for more efficient responses in several sensorimotor control outcome measurements (balance, oculomotor control, head repositioning error) [57]. Since it has been reported that central condition time (N13–N20) and the amplitude of sensorimotor integration (N30) are linearly related to the amount of improvement in FHP and cervical lordosis following an intervention [56], it seems probable that restoration of the sagittal cervical alignment is a primary mechanism for improving the somatosensory system and sensorimotor integration regions, yielding improved sensorimotor control and more efficient motor control output in general. Future studies, however, are needed to clarify this and identify precisely which, if any, specific motor control outcome variables are dependent on and influenced by improved sagittal cervical alignment.

### 4.4. Limitations

By using a matched design, we attempted to adjust for potential confounding characteristics, such as age, sex, BMI, smoking status, marital status, education, and weekly work hours. However, as with any observational study, residual confounding factors, such as the length of the participant’s neck, may exist. N13 is measured from the back of the C6 spinous process and N20 from the scalp. Thus, the latency of the N20 versus the latency of the N13 may be influenced by the length of the neck and the size of the scalp/brain. However, our participant groups were matched for sex and size, so it seems unlikely that neck length would be a significant source of confounding in our populations. Further, any differences in the length of the neck in our two matched groups are likely due to the forward head posture effects on cervical spine kinematics, thus, strengthening our study results [17,18,19]. Still, we recommend that future studies adjust the interpeak latencies to each participant’s neck length.

Additionally, we did not control for certain lifestyle factors, such as physical activity (exercise), and we did not assess the stress or anxiety level experienced by participants, which can affect neural function. Our investigation focused on an asymptomatic population of younger adults; therefore, participants of varying ages and with specific musculo-skeletal disorders should be included in future studies. A further limitation is our method of FHP measurement in that although the CVA is both a reliable and valid quantification method for external FHP [23], the CVA cannot describe the shape and magnitude of the cervical lordotic curve on spine radiographs [45]. Future investigations should use imaging (spine radiographs, MRI, CT) to identify the role that actual vertebral alignment plays in altering sensorimotor integration and somatosensory processing systems. Additionally, we recommend the assessment of patients before and after cervical spine surgical interventions for spine disorders to identify if reductions in FHP to the recommended surgical cutoff values (radiographic FHP < 40 mm) have an effect on improving central conduction time (N13–N20) [5]. Finally, there are several other measurements that represent the sagittal alignment of the head and neck, such as the sagittal head tilt (flexion/extension), sagittal shoulder-C7 angle (protraction/retraction) [58], and these may influence the neurophysiological measures of sensorimotor integration and somatosensory processing. Future investigations should look at more comprehensive measurements of sagittal cervical spine posture in order to confirm, add to, or refute the findings of the current investigation.

### 4.5. Conclusions

Using a matched case-control design in asymptomatic young adults, we identified that forward head posture is associated with differences in central conduction time, sensorimotor integration, and somatosensory processing amplitudes at different neural regions. Full-time work increased the odds of having a higher amplitude of neurophysiological potentials and slower N13–N20 conduction time. Additionally, increases in the CVA (less forward head posture) were found to decrease the amplitudes of somatosensory processing potentials and resulted in a faster N13–N20 conduction time.

## Figures and Tables

**Figure 1 jcm-12-03217-f001:**
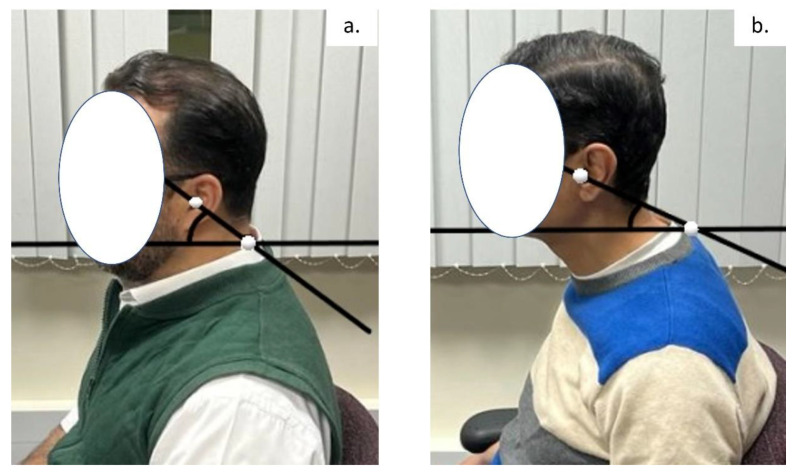
An example of the cranio-vertebral angle (CVA) measurement used with a representative participant with (**a**) normal head posture and (**b**) considerable forward head posture (FHP).

**Figure 2 jcm-12-03217-f002:**
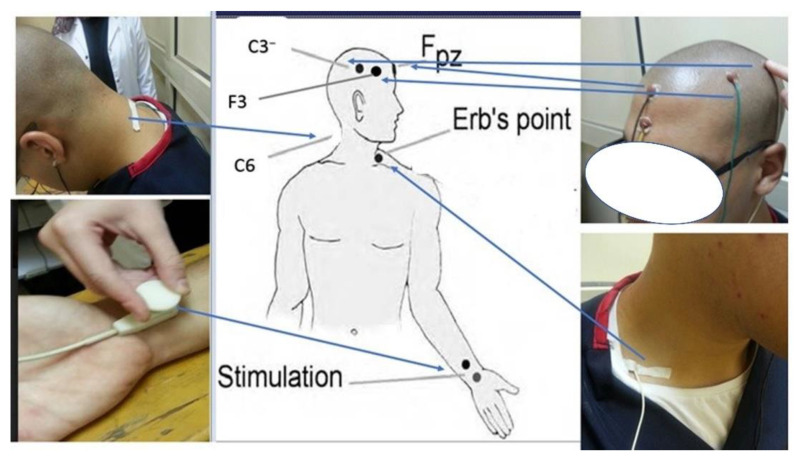
An illustrative example of sensorimotor integration and somatosensory processing measurement.

**Figure 3 jcm-12-03217-f003:**
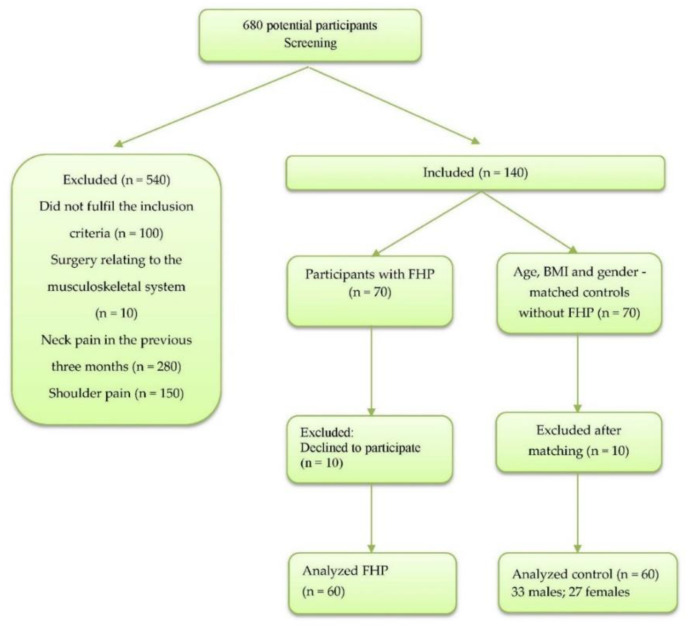
Participants’ inclusion and exclusion flow chart.

**Figure 4 jcm-12-03217-f004:**
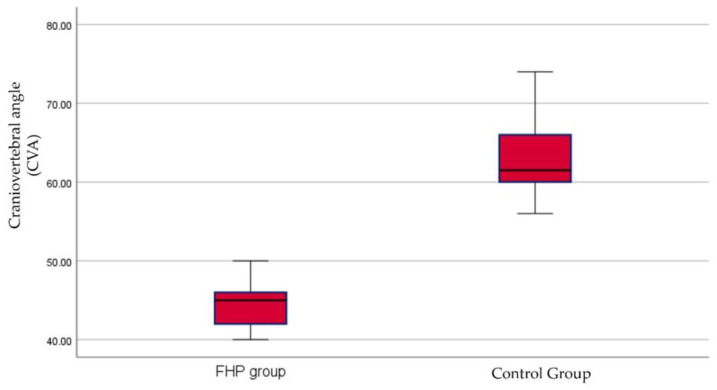
Box and whiskers for craniovertebral angle (CVA) between the forward head posture (FHP) and control groups.

**Figure 5 jcm-12-03217-f005:**
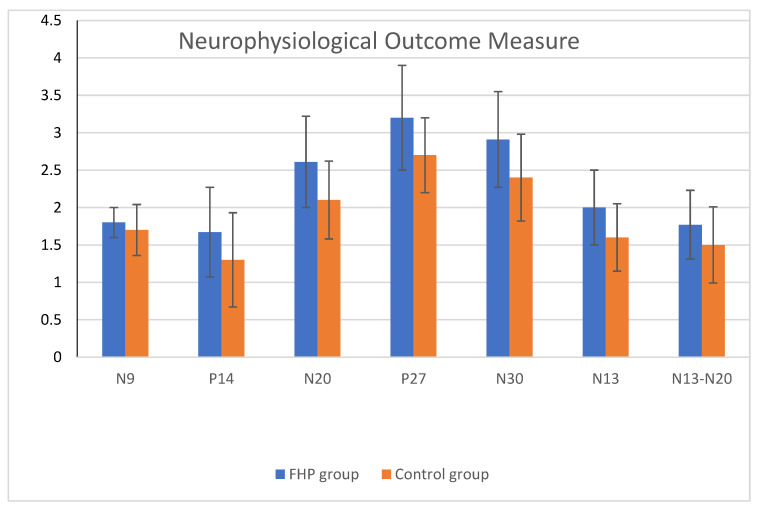
Neurophysiological outcomes for both groups. FHP = forward head posture and Control group = normal head posture group as measured with the CVA. Statistically significant differences between the FHP and control groups for all measured neurophysiological variables were identified, including amplitudes of spinal N13 (*p* < 0.005), brainstem P14 (*p* < 0.005), parietal N20 and P27 (*p* < 0.005), frontal N30 (*p* < 0.005), and N13–N20 interpeak latency as measures of central conduction time (CCT) (*p* = 0.004). There was no significant difference between both groups regarding the amplitudes of the peripheral potential N9 (*p* = 0.07).

**Figure 6 jcm-12-03217-f006:**
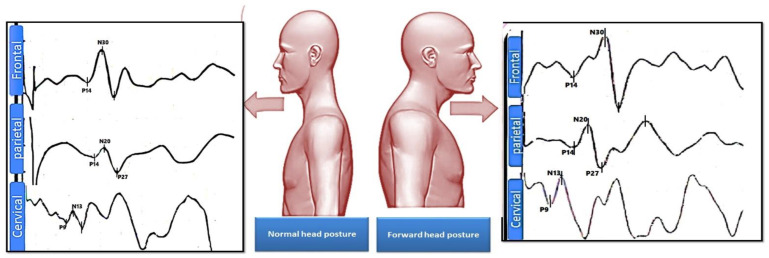
Shown is an example of the frontal N30, parietal N20 and P27, and cervical spinal N13 somatosensory findings for a representative participant with normal head posture on the left side and with considerable forward head posture (FHP) on the right side.

**Table 1 jcm-12-03217-t001:** Descriptive data for demographic variables. No statistically significant differences between the control group (CG) and forward head posture (FHP) groups (*p* > 0.05) were found. The independent *t*-test for continuous data and the Chi-squared test of independence for categorical data were used. Values are presented as mean and standard deviation (SD) for age and weight.

Variable	FHP (n = 60)	CG (n = 60)	*p*-Value
Age (years)	23.5 ± 2	25.9 ± 2	0.07
Weight (kg)	67.2 ± 3	69.2 ± 5	0.11
**Gender (%)**			
Male	35 (58%)	33 (55%)	0.3
Female	25 (42%)	27 (45%)	
**Smoking**			
Light smoker	18	16	0.2
Heavy smoker	0	0	
No Smoker	42	44	
**Educational level**			
Bachelor or Master	43	36	<0.005
High school or less	17	24	
**Marital status**			
Married	32	24	<0.005
Not married	28	36	
**BMI**			
Normal	45	26	<0.005
Obese	15	34	
**Working hours**			
Full-time	22	42	<0.005
Part-time	38	18	

**Table 2 jcm-12-03217-t002:** Differences between the forward head posture group (FHP) and control group (CG) for each outcome measure of the DSSEPs for sensorimotor integration assessment. The amplitudes of the following potentials are reported: peripheral potential N9; spinal N13; brainstem P14; parietal N20 and P27; and frontal N30. Differences in peak latencies between N13 and N20 were measured as central conduction time (N13–N20). CI = confidence interval. (A) is a generalized linear model with adjustment for potential confounding variables, including educational level, marital status, BMI, and number of working hours per week.

Neurophysiological Outcome Measure	FHP Group	Control Group	Mean Difference between the Two Groups	(95% CI)/ Cohen’s d	*p* Value	*p* Value (A)
N9	1.8 ± 0.2	1.7 ± 0.34	0.1	[0.07, 0.21]/0.1	=0.07	0.6
P14	1.67 ± 0.6	1.3 ± 0.63	0.37	[0.25, 0.49]/0.77	<0.005	0.02
N20	2.61 ± 0.61	2.1 ± 0.52	0.51	[0.33, 0.6]/0.9	<0.005	<0.005
P27	3.2 ± 0.7	2.7 ± 0.5	0.5	[0.41, 0.69]/0.8	<0.005	0.04
N30	2.91 ± 0.64	2.4 ± 0.58	0.51	[0.359, 0.69]/2.45	<0.005	0.003
N13	2 ± 0.5	1.6 ± 0.45	0.4	[0.11, 0.35]/0.8	<0.005	0.004
N13–N20	1.77 ± 0.46	1.5 ± 0.51	0.27	[0.07, 0.51]/0.56	=0.004	<0.005

**Table 3 jcm-12-03217-t003:** Correlations (Pearson’s *r*) were used to examine the relationships between the cranial vertebral angle (CVA) in the forward head posture (FHP) group and control group (CG) and the following variables measured: amplitudes of peripheral potential N9; spinal N13; brainstem P14; parietal N20 and P27; and frontal N30 potentials; and central somatosensory conduction time (N13–N20).

Correlation	CVA FHP *r (p*-Value)	CVA CG *r (p*-Value)
N9	−0.44 <0.001	−0.5 <0.001
N13	−0.67 <0.001	−0.54 <0.001
P14	−0.58 <0.001	−0.57 <0.001
N20	−0.49 <0.001	−0.51 <0.001
P27	−0.58 <0.001	−0.6 <0.001
N30	−0.64 <0.001	−0.61 <0.001
N13–N20	−0.61 <0.001	−0.56 <0.001

**Table 4 jcm-12-03217-t004:** Logistic regression models showing the predictors of the neurophysiological outcomes.

	P14	N20	P27	N30	N13	N13–N20
Predictors	Odds ratios (*p*-value)	Odds ratios (*p*-value)	Odds ratios (*p*-value)	Odds ratios (*p*-value)	Odds ratios (*p*-value)	Odds ratios (*p*-value)
BMI (Obesity)	0.4 0.06	0.23 0.06	0.13 0.3	0.16 0.34	0.2 0.06	0.2 0.06
Educational level (Bachelor or Master)	1.2 0.4	3.2 0.08	2.3 0.3	1.2 0.4	2.4 0.32	1.5 0.42
Marital status (Not married)	1.54 0.2	1.54 0.2	1.3 0.3	1.3 0.3	1.5 0.2	1.8 0.09
Weekly working hours (Full-time)	13.1 <0.005	12.4 <0.005	19.5 <0.005	25.9 <0.005	28 <0.005	19.4 <0.005
CVA	0.41 <0.005	0.3 <0.005	0.3 <0.005	0.57 <0.005	0.23 <0.005	0.34 <0.005

## Data Availability

The datasets analyzed in the current study are available from the corresponding author upon reasonable request.

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
