# Peer review of "Does Forward Head Posture Influence Somatosensory Evoked Potentials and Somatosensory Processing in Asymptomatic Young Adults?"

_jcm, 2023, doi:10.3390/jcm12093217_

Round 1

Reviewer 1 Report

Abstract:

-       You said "We assessed different neural structures of the somatosensory pathway…" the word Structures is not appropriate.

-       You said "Pearson correlation was used to evaluate the relationship between FHP and sensorimotor integration and SEP measurements." I think you meant craniovertebral angle here.

-       You said "The CVA significantly correlated with all measured variables (p< .001)". I would like to see the Correlation ratio as the indication of the correlation's strength.

-       You finished your abstract with the result and the future study suggestion. I would like to see  briefly the clinical impact of your result as well.

Method:

-       I can see that you cases were young. Was age in your inclusion criteria list? If it is so, please add it to your inclusion criteria. Since the age may reflects the chronicity of the FHP and the possible degenerative changes in the spine, it is an important matter.

-       Line 133, the title "Outcome Measures", outcome is not appropriate word, considering that you did not have any interventions, "measurement techniques" is a better title here.

Results:

Line 282- 284: under the title "Correlation of findings between groups" it appears that you have measured the correlation in each group separately. Why? Since both groups have no symptoms and pain free, there was no other factors except the CVA that can affect the evoked potentials. Integration of both groups could have given you a wider range of CVA for correlation assessment.

Discussion:

Does the result of your study have clinical implication? Please add a paragraph and discuss it.

Reviewer 2 Report

I found this article really interesting. Evaluation of FHP would have been better with xrays of course but i think it is fine for a first evaluation. I would also mention in discussion that reduction of FHP in surgical patients will need to be assessed regarding central conduction 

Reviewer 3 Report

A very difficult article to read. The intro is very long and doesn't add much.  It needs to be edited and shortened.  The purpose of the study should be clearly indicated.  The selection of groups is not entirely clear.  Research methodology and statistics at an average level, discussion to be improved, hard to find what it refers to and for what purpose.  Limitations too broadly described.  The conclusion is too long and not clear and does not reflect the conclusions of the entire study.

Round 2

Reviewer 3 Report

Corrections have been added. The paper is more adequate. The final decision should be made by the Editor.